# Influence of Prolonged Visual Display Terminal Use on Physical and Mental Conditions among Health Care Workers at Tertiary Hospitals, Taiwan

**DOI:** 10.3390/ijerph19073770

**Published:** 2022-03-22

**Authors:** Meng-Ting Tsou

**Affiliations:** 1Department of Family Medicine, MacKay Memorial Hospital, Taipei City 10449, Taiwan; mttsou@mmh.org.tw or mttsou@gmail.com; Tel.: +886-2-2543-3535 (ext. 2131/2132); 2Department of Occupation Medicine, MacKay Memorial Hospital, Taipei City 10449, Taiwan; 3MacKay Junior College of Medicine, Nursing, and Management, New Taipei City 25245, Taiwan

**Keywords:** healthcare workers, VDT use, CVS, tertiary hospital

## Abstract

This study aimed to examine the effects of prolonged VDT working time on physical and mental health disadvantages among health care workers (HCWs) in tertiary hospitals based on their work characteristics, age, and sex. Included in the study were 945 and 1868 participants in the non-doctor/nurse and doctor/nurse groups, respectively. The questionnaire included VDT usage-related information, the Nordic Musculoskeletal Questionnaire (NMQ), computer vision syndrome (CVS), perceived occupational stress, burnout, the Brief Symptom Rating Scale-5 (BSRS-5), and self-rated health (SRH). After adjustment, multiple logistical regression analysis revealed that the two groups showed that the longer the VDT working time, the higher the risk of muscle pain, severe headaches, severe job stress, and self-assessed bad sleep quality. This showed that the condition of the doctor/nurse group was more severe than that of the non-doctor/nurse group. According to the stratified analysis by sex and age, in the group of women under the age of 30, the adjusted odds ratio value of physical and mental conditions increased with longer VDT working time and was statistically significant. The result show that it is important to reduce daily VDT exposure for doctor, nurses, and women under 30.

## 1. Introduction

Visual display terminals (VDTs) have been used in almost all institutions and organizations. In Taiwan, the VDT usage rate in domestic private enterprises is approximately’ 75%, and that of public organizations is as high as 100% [1]. According to national surveys, as much as 77.7% of the general population in Taiwan use VDTs. A total of 74.2% of workers need to use VDTs at work. Among them, the usage rate among professionals (including health care workers [HCWs], teachers, and other occupations) is as high as 82.7%. Among the workers who use VDTs, 72.5% were men, while 69.3% were women [2]. In Taiwan, as early as 2011, the Government Institute of Labor, Occupational Safety, and Health, Ministry of Labor, highlighted the damage to the musculoskeletal system caused by VDT use [3]. Due to more significant medical pressure, excessive working hours and shift work have caused many health problems for HCWs in tertiary hospitals [4,5], thus requiring evaluation of VDT use.

VDTs are used in many workplaces, and their use may increase the risk of visual, musculoskeletal, and mental problems [6,7]. Work-related musculoskeletal disorders (MSDs) are a serious problem among HCWs. We are concerned about these disorders because they can be severely debilitating [8]. MSDs are associated with high costs to HCWs, such as absenteeism, lost work productivity, increased self-health care, and disability; MSD patient safety cases are more severe than the average nonfatal injury or illness [8]. The etiology of MSDs, especially on neck pain, which has been discussed in previous studies, including personal factors (e.g., sex and age) [7,9], work-related factors (e.g., repetitive work, sedentary and static posture), as well as psychosocial factors, are considered to be risk factors for MSDs in office workers [10,11,12].

Previous studies have revealed the duration of VDT use at work and its association with physical and mental symptoms [13]. Cheng et al. found that when the 944 Internet employees’ daily VDT operation time exceeded 11 h (including work and leisure time), it induced back pain (odds ratio [OR] = 3.59), wrist pain (OR = 1.88), hip pain (OR = 2.42), dry eyes (OR = 2.22), and eye pain (OR = 2.16). In addition, the risk of severe occupational stress (OR = 6.75) and job burnout (OR = 2.66) increased [13]. People who spend more than four hours using VDTs are at a higher risk of developing depression [14]. It has also been reported that people with depressive symptoms spend significantly more time using VDTs per day [14].

Many studies have focused on the relationship between VDT working time and the physical and mental health of people of different ages and occupations, or the general population, including depression and insomnia [15,16]. However, HCWs employed in tertiary hospitals are rarely discussed. The physical and mental health of HCWs will affect their work quality; therefore, this study focuses explicitly on the VDT working time of HCWs to help protect their physical and mental health.

## 2. Materials and Methods

### 2.1. Study Population

This cross-sectional study was conducted at the Occupation Safety Department of MacKay Memorial Hospital, a 2000-bed tertiary teaching center, in both the Taipei and New Taipei branches, in Taiwan. Questionnaires were administered between December 2018 and March 2019. According to previous research [17,18], doctors and nurses have similar results in many fields, such as direct patient care, job stress, shift work, working hours, and burnout. For these reasons, we divided HCWs into two groups in this study, namely, doctor/nurse and non-doctor/nurse groups. The non-doctor/nurse group included technicians, administrative staff, pharmacists, radiologists, nutritionists, and others. A total of 3038 doctor/nurse and 1796 non-doctor/nurse participants were invited, and 1868 and 945 participants from the two groups completed the questionnaires. The response rates were 62.1% (1868/3013) in the doctor/nurse group and 53.1% (945/1780) in the non-doctor/nurse group. A sample size of 2533 achieved 90% power using the two-tailed test (OR = 1.59; probability of null hypothesis = 0.15; alpha error = 0.05; power = 0.9; R^2^ for other confounding factors = 0.5) [19].

### 2.2. Study Instrument

Information was obtained using structured questionnaires, which were designed according to the Institute of Labor, Occupational Safety, and Health, Ministry of Labor (Survey of Perceptions of Safety and Health in the Work Environment in 2016 Taiwan; ILOSH105-M309) [20]. The following data were collected: socioeconomic status, working characteristics, VDT usage-related information, job burnout syndrome, and physical and mental health. Analysis of the internal consistency of each part of the scale showed the Cronbach’s α to be between 0.82 and 0.93, indicating good reliability [20].

#### 2.2.1. Demographic Data

The sociodemographic characteristics included age, sex, and education level. Personal habits included smoking (current or past/never), alcohol consumption (0–1 drink per week/≥2 drinks per week), and exercise (≥30 min/time) frequency (nerve, once per week, 2–4 times per week, ≥5 times per week) [20].

#### 2.2.2. Work Characteristics

Basic work issues include department and seniority. Information about working hours and shift jobs was derived from the work status provided by participants during the month before the survey. The participants were categorized into regular class, night shift, and three shifts. The regular class was defined as starting work in the morning with no rotation. Work shifts that started in the morning, afternoon, or evening and rotation were classified into three shifts. However, if they extended to midnight they were classified as night shifts [21].

According to the Labor Standards Law in Taiwan, Working Hours Standards for Nursing Staffs in Medical Institutions, the regular working hours per day shall not exceed 8 h. Furthermore, the total working hours per week shall not exceed 48 h [22]. Based on national regulations and South Korean research, we divide weekly working hours into four grades: ≤45 h, 46–50 h, 51–60 h, and ≥61 h [23].

#### 2.2.3. VDT Usage-Related Information

Daily VDT work time was assessed using an open question: “On average, how many hours do you spend with your screen on work on each working day (including overtime)?” Three options were provided: <2 h 2–6 h, and ≥6 h [12].

### 2.3. Physical Condition

Questions on MSDs were adapted from the Nordic Musculoskeletal Questionnaire (NMQ) [24], which is the most common outcome measurement. Some symptoms specific to VDT workers were selected, such as aches, pain, or discomfort in the neck, shoulders, elbows/forearms, wrists/hands, upper and low back, and hip or leg regions. Specific body pain was queried using the question, “Have you experienced pain or discomfort of the following body parts in the past six months (yes/no)?” If the participant stated that the musculoskeletal symptoms resulted from a non-work-related accident/trauma (e.g., sports or traffic accident), they were be excluded from the study.

Computer vision syndrome (CVS), a phenomenon associated with the use of VDTs, has symptoms of eyestrain, headaches, dry eyes, diplopia, and blurred vision, and is also referred to as asthenopia [25]. The first component involves external sensations associated with dry eyes (e.g., burning and irritation), while the second involves internal symptoms (e.g., headaches). Eye discomfort includes reduced vision, dry eyes, ocular soreness, and eye irritation [25]. Eye discomfort and headaches were categorized as never, mild to moderate, or severe. The severity of CVS was categorized as never (no headaches and eye discomfort), mild to moderate (mild to moderate headaches and mild to moderate eye discomfort), or severe (severe headaches and severe eye discomfort).

### 2.4. Mental Condition

Perceived occupational stress was evaluated on a scale from 0 (no pressure) to 10 (greatest pressure), where respondents selected a number to suggest their general occupational stress. The scale was categorized as 0–3 (mild), 4–6 (moderate), and 7–10 (severe).

Burnout domains were evaluated using the Chinese version of the Maslach Burnout Inventory—Health Services Survey test (MBI-HSS) according to Lu et al. (Cronbach’s α, 0.84) [26]. Three domains of the MBI-HSS were evaluated: emotional exhaustion (EE), depersonalization (DP), and personal accomplishment (PA). The MBI-HSS includes a total of 22 questions, and occurrence frequency is scored using a seven-point Likert scale from 0 (never felt) to 6 (felt every day) [26]. EE scores ≥ 27, DP scores ≥ 13, and PA scores ≤ 31 were categorized as high job burnout [24]. The EE and DP scores were directly related to burnout, while the PA scores showed a negative association [26]. The internal consistency coefficients (Cronbach’s α) of the three dimensions of EE, DP, and PA were 0.77, 0.83, and 0.82, respectively [26].

Mental health was assessed using the BSRS-5, which is a self-assessment questionnaire that includes five items. It requires respondents to report whether they felt tense, downcast, irritated, or inferior, or had any problems falling asleep in the past week. Responses are rated on a scale of 0–4, with 0 being “nothing” and four being “extremely”. The total scores ranged from 0 to 20 (Cronbach’s α coefficient: 0.77–0.90) [27] and a score of ≥6 was used as the cutoff point for psychiatric cases. The BSRS-5 was divided into four groups: “no symptoms” (0–5), “mild” (6–9), “moderate” (10–14), and “severe” (>15) [27]. The BSRS-5 accurate classification rate was 76.3% (sensitivity: 78.9%; specificity: 74.3%) [27].

Self-rated health (SRH) was assessed using two questions. The self-assessment health status question, “In general, how is your health?”, had three possible answers: “good”, “moderate”, or “poor” and the self-assessment sleep quality question, “How about your sleep quality?”, had two possible answers: “good” and “poor” [28,29]. The severity of SRH was categorized as poor (poor self-assessment health status + poor self-assessment sleep quality), and good (good or moderate self-assessment health status + good self-assessment sleep quality.

### 2.5. Ethics Approval

The study protocol was evaluated and approved by the Human Research Ethics Committee of Mackay Memorial Hospital (project research number 18MMHISO150). All participants provided written informed consent. Data confidentiality was preserved, considering ethical issues such as autonomy and respect for people. Moreover, all the guidelines of the Declaration of Helsinki were followed. To ensure data confidentiality, participant identification information was replaced with a folio number.

### 2.6. Statistical Analysis

A descriptive analysis was performed to characterize the population sample. Data are presented as means ± standard deviations for continuous variables, and numbers and percentages are presented for categorical variables. VDT use at work was classified into three categories (<2 h, 2–6 h, and ≥6 h), and frequencies and percentages were calculated for each factor. For comparisons between the groups, a Student’s *t*-test was used to analyze continuous variables and a chi-square test was used for categorical variables. Multivariate logistic regression analyses were performed to investigate the possible associations between VDT working time and physical/mental conditions after adjusting for age, sex, personal habits, and work characteristics. Finally, we evaluated the association between the number of pain areas (≥7/<7), CVS (yes/no), BSRS-5 (≥6/<6), burnout (yes/no), and SRH (poor/not poor) in the strata of age and sex using logistic regression analysis. We grouped according to median age: 45 years in the non-doctor/nurse group, 28 years in the doctor/nurse group, 32 years in all groups, and 30 years as the cut-off point for the estimated stratification. All analyses were performed using SPSS 22 (IBM Corp., Armonk, NY, USA) for Windows. Two-sided *p* values of <0.05 were considered statistically significant.

## 3. Results

### 3.1. Characteristics of the Participants According to Profession with VDT Working Time

Table 1 shows 945 participants in the non-doctor/nurse group and 1868 participants in the doctor/nurse group. Participants in the non-doctor/nurse group used VDTs at work for less than two hours (400/945, 42.3%), while those in the doctor/nurse group used them for more than six hours (51.1%). The age of the non-doctor/nurse group with VDT working times of less than two hours was the oldest among the three groups (non-doctor/nurse, <2 h: 45.9 ± 11.98 Y/O; doctor/nurse, ≥6 h: 36.91 ± 10.37 Y/O). Regardless of the group, it was found that the group that had longer daily VDT working time (≥6 h) had a higher proportion of senior seniority and longer working hours. In the doctor/nurse group, it was found that participants with longer daily VDT work times had a higher proportion of night and three shifts. In addition, the frequency of smoking and fewer exercise habits was higher in the group with VDT working times of less than two hours.

### 3.2. Physical and Emotional Symptoms of the Participants According to Profession with VDT Working Hours

Table 2 shows the proportion of physical and mental symptoms in groups with different VDT working times. In the non-doctor/nurse group, with the increase in VDT working time, neck, shoulder, back, knee, and ankle pain increased. In the doctor/nurse group, pain increased in all parts, and the total number of painful parts also increased. (<2 h: 2–6 h: ≥6 h = 3.16 ± 2.5: 3.92 ± 2.78: 4.32 ± 2.60, *p* < 0.001). Regardless of the group, severe eye discomfort, severe headaches, and moderate to severe CVS increased with an increase in VDT working time. The mental conditions of the two groups worsened with the increase in VDT working time. Moreover, the proportion of BSRS-5 scores ≥ 6, burnout, moderate to severe job stress, and bad SRH all increased at the same time.

### 3.3. Adjusted Odds Ratio (aOR) of Physical and Mental Symptoms among HCWs Varies According to Profession with Different VDT Working Times

Table 3 shows the results of the multiple logistical regression used to analyze the aOR change in physical and mental symptoms caused by the length of VDT working time in non-doctor/nurse and doctor/nurse groups. Model 1 is unadjusted, and model 2 is adjusted by age, sex, smoking, drinking, exercise, seniority, working hours per week, and working style. Table 3a shows the status of non-doctor/nurse staff. After adjustment, the longer the VDT working time, the more the aOR of neck, shoulder, knee, and ankle pain increased, and this relationship was statistically significant. In addition, the risk of eye discomfort, headaches, severe job stress, and self-assessed bad sleep quality increased with longer VDT working times. However, there were no statistical differences in CVS, the BSRS-5 score, burnout, self-assessed health status, and sleep time. Table 3b shows that the condition of the doctor/nurse staff was more severe than that of non-doctor/nurse staff. The adjusted data show that the longer the VDT working time, the higher the aOR value of physical and mental symptoms, and this effect was statistically significant. The only exception was that CVS and the BSRS-5 score did not reach statistical significance.

### 3.4. Analysis of Physical and Emotional Symptom Prevalence by Stratified Age and Sex Subgroups among HCWs According to Different VDT Working Times

Table 4 shows the results of the analysis stratified by sex and age. Only males older than 30 appeared at higher risk of CVS in the VDT working time ≥ six hours group compared with the < two hours group [OR = 2.94 (95% CI = 1.09–3.98)]. The others, including physical and mental symptoms, did not reach statistical significance. In the group of females younger than 30, no matter the number of pain areas, CVS, the BSRS-5 score, burnout, and SRH increased with VDT working times; the aOR values increased, and they were statistically significant. In women older than 30, the number of pain areas and burnout did not increase the aOR value with the increase in VDT working times. However, the aOR values of CVS, the BSRS-5 score, and SRH increased with VDT working times and were statistically significant.

### 3.5. Characteristics of the Participants According to Profession

The analysis was performed in the doctor/nurse and non-doctor/nurse groups and is shown in the Appendix A. Appendix A lists the basic characteristics. Compared with the non-doctor/nurse group, the doctor/nurse group is younger, has more female staff, a higher proportion of college degrees or above, seniority of less than two years, and more than 46 h of night shifts. Smoking and exercise habits were higher in the non-doctor/nurse group. Appendix A shows the physical symptoms, and Appendix A shows the emotional stress symptoms. The proportion of both was significantly higher in the doctor/nurse group than in the non-doctor/nurse group.

## 4. Discussion

This study found that prolonged VDT working time (≥6 h) significantly increased the risk of neck and shoulder pain, eye discomfort (including dryness and pain), headaches, depression, burnout, high job stress, and poor SRH. The impact on the doctor/nurse group was more significant than on the non-doctor/nurse group. After stratifying by age and gender, it was found that the impact on female HCWs was more significant than on males, especially in workers younger than 30.

In our study, 51.1% (955/1868) of the staff in the doctor/nurse group had an average VDT working time of more than six hours per day; in contrast, staff in the non-doctor/nurse group had an average of less than two hours per day (42.3%, 400/745). Since there were two distinct groups according to VDT working time, this study separated the two groups for analysis.

The 2016 national random VDT usage survey was conducted with 23,465 Taiwanese; of whom about 77.7% reported personal use of VDTs, including 78.9% and 76.6% among men and women, respectively. According to the age-stratified analysis, it was found that the 20–31 age group had the highest usage, as high as 98.7%. The VDT usage rate gradually decreased as age increased [2]. In our study, we also found VDT working times of more than six hours for women under the age of 30 had a greater impact on physical and mental health.

Ye et al. showed that among 3070 administrative office employees aged 18 to 67 (average age: 39.9), 76% of whom were VDT users, the duration of the daily use of VDTs was related to eye fatigue, neck or upper limb pain, back pain, and psychological distress [30]. The major occupational risk causing neck pain for HCWs is prolonged awkward postures such as bending over a desk for hours, placing VDTs too high or too low, and so forth [8].

Previous studies showed that visual fatigue was significantly associated with perceived anxiety (OR 7.40; 95% CI 1.77–31.3), psychosocial factors (OR 1.03; 95% CI 1.01–1.07), and VDT usage time (OR 1.27; CI 95% 1.04–1.53) in a dose-related manner [31]. A systematic review and meta-analysis of studies comparing those who reported more or less screen time-based sedentary behavior (ST-SB), showed that persons reporting more ST-SB had a significantly higher risk of depression, especially in female groups (pooled OR: 1.18) [14,32]. More time spent using a computer on weekdays was associated with an increased risk of anxiety and depression [33]. Conversely, another study showed that the evidence for anxiety symptoms was more limited than that for depressive symptoms in adults [33]. According to self-reported depressive and anxiety disorder in a nationally representative sample of South Korean workers, after adjustment factors, the OR for depressive and anxiety disorder was higher in workers using computers for more than 75% of their workday (OR 1.69, 95% CI 1.30−2.20) than in workers using computers for less than 50% of their shift [23]. Burnout appeared to be a significant contributing factor to the intensity and frequency of VSD [34].

Giahi et al., collected data from 382 Iranian bank tellers, and found significantly higher insomnia symptom scores, higher job stress, and lower job satisfaction among participants with more than six hours of daily VDT use after adjusting for multiple confounding factors (*p* < 0.001) [6]. Yoshioka et al. investigated 2417 Japanese clerks (men: 2030; women: 387) at local government offices. They found that VDT work for six hours or more per day was significantly associated with insomnia (OR = 1.62; 95% CI: 1.16–2.27), even after adjusting for possible confounding factors [35].

Regarding the impact of VDT job and working times on physical and mental health, many previous studies have focused on sex analysis. Madhav et al. showed that moderate or severe depression levels were associated with higher VDT usage times (>6 h/day) (aOR: 2.3; 95% CI: 1.602–3.442). VDT usage time can predict depression levels among adults, especially women [16]. A systematic review and meta-analysis included a total of 11,365 VDT workers and showed that the prevalence of dry eye disease (DED) was more frequent in females than in males and increased with age [36].

In the Osaka study of 2013, 561 young and middle-aged Japanese office workers completed a questionnaire. The results showed that 76.5% of women had DED, which was a higher rate than that among men. Workers over the age of 30 had a higher risk of DED, as did workers using VDTs for more than eight hours per day [37]. Our study found a higher OR of CVS for women over the age of 30 and for men under the age of 30, especially in women with more than six hours of VDT working time. Kim et al. surveyed 2178 bank workers (62.8% male, 37.2% female) using a self-administered structured questionnaire consisting of 30 fatigue symptoms. Women had significantly higher scores than men (*p* < 0.001). In multiple regression analysis, the female sex, more VDT work hours, overtime work, and a young age were significantly associated with high self-rated health scores [38].

The most effective way to eliminate MSDs, CVS hazards, and mental health problems in the tertiary hospital workplace is to develop health programs. The programs should be aimed at three fields: first, advocating healthy lifestyle behaviors (stretching exercise, quitting smoking, and adequate sleep time/quality) [23,39]. Second, raising workers’ awareness of the ergonomics of their workstation (seat height adjustment, working desktop height, keyboard and mouse arrangement, screen contrast and brightness, and sight distance). Third, work organization (VTD working duration, rotation and shift job, job stress, and social support) [23,39,40], especially for women and younger workers.

Several limitations should be considered when interpreting our results. First, there was a significant difference in the sample size between men (302, 10.7%) and women (2511, 89.3%), mainly because of the number of female nursing staff. There were some restrictions for men when stratifying the analysis by age and gender because of the small sample size. Second, all participants were from different branches of the two tertiary hospitals. Therefore, the study results may not be generalizable to other HCWs. Third, because this was a cross-sectional study, the causal nature of the association between VDT working time and physical or mental conditions could not be determined. Fourth, relevant information about VDT working time was obtained through self-answered questionnaires, which are not always reliable and could have caused reporting bias.

## 5. Conclusions

Prevention of MSDs, visual symptoms, and mental health in hospitals an the organizational level consists of workplace risk assessment. The modifiable individual and work-related factors were as follows: improving health status, job satisfaction, reduction of working hours on VDTs, avoiding prolonged sitting and static postures, resting time during work hours, and performing regular daily exercises [7,8,9,10,23,39,40]. Thus, this research can help to protect HCWs in tertiary hospitals from the adverse effects associated with VDT work [6].

## Figures and Tables

**Table 1 ijerph-19-03770-t001:** Demography and characteristics of personal habits and work among different daily VDT working durations in hospital employees.

Variables	Non-Doctor/Nurse									*p* Value	Cochran-Armitage Trend Test	Doctor/Nurse									*p* Value	Cochran-Armitage Trend Test
	VDT Working Time (945)											VDT Working Time (1868)										
	<2 h			2–6 h			≥6 h					<2 h			2–6 h			≥6 h				
	(*n* = 400, 42.3%)			(*n* = 207, 21.9%)			(*n* = 338, 35.8%)					(*n* = 503, 26.9%)			(*n* = 410, 22.0%)			(*n* = 955, 51.1%)				
**Age, years**	45.90	±	11.98	43.83	±	10.50	43.75	±	10.53	0.02	0.01	31.09	±	10.90	36.65	±	11.06	36.91	±	10.37	<0.001	<0.001
**Women, *n* (%)**	318		79.50%	167		80.68%	286		84.62%	0.19	0.08	416		83.53%	388		94.63%	936		98.01%	<0.001	<0.001
**Working character**																						
Seniority, *n* (%)										<0.001	<0.001										<0.001	<0.001
<2 years	132		33.00%	8		3.86%	19		5.64%			278		55.27%	34		8.29%	58		6.07%		
2–4 years	67		16.75%	43		20.77%	54		16.02%			77		15.31%	84		20.49%	146		15.29%		
4–10 years	88		22.00%	68		32.85%	87		25.82%			73		14.51%	133		32.44%	360		37.70%		
>10 years	113		28.25%	88		42.51%	177		52.52%			75		14.91%	159		38.78%	391		40.94%		
Working hours/week, *n* (%)										0.28	0.56										<0.001	0.001
≤45 h	302		75.50%	173		83.57%	265		78.64%			270		53.68%	258		62.93%	530		55.50%		
46–50 h	91		22.75%	30		14.49%	64		18.99%			145		28.83%	129		31.46%	335		35.08%		
51–60 h	6		1.50%	3		1.45%	5		1.48%			32		6.36%	22		5.37%	61		6.39%		
≥61 h	1		0.25%	1		0.48%	3		0.89%			56		11.13%	1		0.24%	29		3.04%		
Work style, *n* (%)										<0.001	<0.001										<0.001	0.74
Regular class	244		61.00%	139		67.15%	309		91.42%			222		44.14%	150		36.59%	381		39.90%		
Night shift	8		2.00%	6		2.90%	3		0.89%			11		2.19%	31		7.56%	74		7.75%		
Three shifts	148		37.00%	62		29.95%	26		7.69%			270		53.68%	229		55.85%	500		52.36%		
**Personal habits**																						
Smoking, *n* (%)	30		7.50%	11		5.31%	8		2.37%	0.01	0.002	22		4.37%	17		4.15%	15		1.57%	0.002	0.001
Drinking, *n* (%)	27		6.75%	19		9.18%	18		5.33%	0.22	0.48	43		8.55%	26		6.34%	54		5.65%	0.10	0.04
Exercise frequency (time/week), *n* (%)										0.004	0.02										0.14	0.03
never	162		40.50%	98		47.34%	113		33.43%			252		50.10%	189		46.10%	518		54.24%		
1 time per week	122		30.50%	59		28.50%	95		28.11%			127		25.25%	118		28.78%	238		24.92%		
2–4 times per week	97		24.25%	47		22.71%	113		33.43%			109		21.67%	91		22.20%	180		18.85%		
≥5 times per week	19		4.75%	3		1.45%	17		5.03%			15		2.98%	12		2.93%	19		1.99%		

**Table 2 ijerph-19-03770-t002:** Prevalence of physical health conditions among different durations of daily VDT working hours in hospital employees.

Variables	Non-Doctor/Nurse									*p* Value	Cochran-Armitage Trend Test	Doctor/Nurse									*p* Value	Cochran-Armitage Trend Test
	VDT Working Time											VDT Working Time										
	<2 h			2–6 h			≥6 h					<2 h			2–6 h			≥6 h				
	(*n* = 400)			(*n* = 207)			(*n* = 338)					(*n* = 503)			(*n* = 410)			(*n* = 955)				
**Musculoskeletal system**																						
Musculoskeletal pain, *n* (%)																						
Neck	164		41.00%	125		60.39%	225		66.57%	<0.001	<0.001	300		59.64%	258		62.93%	686		71.83%	<0.001	<0.001
Shoulders	226		56.50%	148		71.50%	255		75.44%	<0.001	<0.001	322		64.02%	302		73.66%	789		82.62%	<0.001	<0.001
Back	141		35.25%	91		43.96%	151		44.67%	0.02	0.01	160		31.81%	171		41.71%	453		47.43%	<0.001	<0.001
Elbows	109		27.25%	57		27.54%	94		27.81%	0.99	0.86	68		13.52%	99		24.15%	225		23.56%	<0.001	<0.001
Lower back or waist	214		53.50%	118		57.00%	179		52.96%	0.63	0.91	186		46.50%	89		43.00%	159		47.04%	<0.001	<0.001
Wrists or hands	159		39.75%	89		43.00%	143		42.31%	0.68	0.47	138		27.44%	160		39.02%	404		42.30%	<0.001	<0.001
Hips or legs	88		22.00%	43		20.77%	79		23.37%	0.77	0.67	105		20.87%	104		25.37%	283		29.63%	0.001	<0.001
Number of pain areas	2.70	±	2.31	3.05	±	2.20	2.99	±	2.02	0.09	0.07	2.46	±	1.78	3.06	±	2.16	3.36	±	2.01	<0.001	<0.001
**Visual system**																						
Eye discomfort, *n* (%)										<0.001	<0.001										<0.001	<0.001
Nerve	218		54.50%	78		37.68%	79		23.37%			254		50.50%	150		36.59%	237		24.82%		
Mild-moderate	151		37.75%	107		51.69%	174		51.48%			210		41.75%	219		53.41%	495		51.83%		
Severe	31		7.75%	22		10.63%	85		25.15%			39		7.75%	41		10.00%	223		23.35%		
Headache, *n* (%)										<0.001	<0.001										<0.001	<0.001
Nerve	218		54.50%	78		37.68%	79		23.37%			254		50.50%	150		36.59%	237		24.82%		
Mild-moderate	151		37.75%	107		51.69%	174		51.48%			210		41.75%	219		53.41%	495		51.83%		
Severe	31		7.75%	22		10.63%	85		25.15%			39		7.75%	41		10.00%	223		23.35%		
CVS, *n* (%)										<0.001	<0.001										<0.001	<0.001
Nerve	218		54.50%	78		37.68%	79		23.37%			254		50.50%	150		36.59%	237		24.82%		
Mild-moderate	58		14.50%	41		19.81%	88		26.04%			93		18.49%	60		14.63%	172		18.01%		
Severe	124		31.00%	88		42.51%	171		50.59%			156		31.01%	200		48.78%	546		57.17%		
**Mental health condition**																						
BSRS-5 score (mean ± SD)	3.65	±	3.32	4.37	±	3.17	4.62	±	3.37	<0.001	<0.001	5.21	±	4.00	5.46	±	3.73	5.90	±	3.92	0.004	0.001
BSRS-5, *n* (%)										0.46	0.02										0.02	0.03
≤5	313		78.25%	153		73.91%	240		71.01%			319		63.42%	249		60.73%	531		55.60%		
6–9	61		15.25%	37		17.87%	64		18.93%			100		19.88%	98		23.90%	258		27.02%		
10–14	23		5.75%	15		7.25%	30		8.88%			69		13.72%	55		13.41%	128		13.40%		
≥15	3		0.75%	2		0.97%	4		1.18%			15		2.98%	8		1.95%	38		3.98%		
Burnout, *n* (%)	28		7.00%	9		4.35%	24		7.10%	0.38	0.99	21		4.17%	22		5.37%	79		8.27%	0.01	0.002
Job stress, *n* (%)										0.004	0.001										<0.001	
Mild	307		76.75%	157		75.85%	218		64.50%			308		61.23%	234		57.07%	378		39.6%		
Moderate	50		12.50%	26		12.56%	64		18.93%			83		16.50%	72		17.56%	190		19.9%		
Severe	43		10.75%	24		11.59%	56		16.57%			112		22.27%	104		25.37%	387		40.5%		
**SRH**																						
Self-assessment health status, *n* (%)										0.004	0.001										<0.001	<0.001
Good	165		41.25%	65		31.40%	96		28.40%			190		37.77%	111		27.07%	200		20.94%		
Moderate	211		52.75%	131		63.29%	218		64.50%			279		55.47%	262		63.90%	619		64.82%		
Poor	24		6.00%	11		5.31%	24		7.10%			34		6.76%	37		9.02%	136		14.24%		
Sleep duration, hs (mean ± SD)	6.60	±	1.01	6.57	±	1.03	6.49	±	0.94	0.33	0.14	6.64	±	1.01	6.63	±	1.12	6.55	±	1.12	0.23	0.13
Self-assessment of sleep quality										0.01	0.004										<0.001	0.001
Good	200		50.00%	82		39.61%	134		39.64%			206		41.04%	180		43.90%	315		33.02%		
Poor	200		50.00%	125		60.39%	204		60.36%			296		58.96%	230		56.10%	639		66.98%		

Note: *n* = numbers, SD = standard deviation, VDT = visual displayer terminal, BSRS = Brief Symptom Rating Scale, CVS = computer vision syndrome, SRH = self-rated health.

**Table 3 ijerph-19-03770-t003:** Adjusted odds ratio (aOR) and 95% CI of the health condition according to profession with different durations of daily VDT working time.

a. Non-Doctor/Nurse Group																
Variables			Model 1							Model 2						
			2–6 h			≥6 h				2–6 h			≥6 h			
			aOR	95% CI	*p* Value	aOR	95% CI	*p* Value	Cochran-Armitage Trend Test	aOR	95% CI	*p* Value	aOR	95% CI	*p* Value	Cochran-Armitage Trend Test
**Musculoskeletal system**																
Neck			2.19	(1.56–3.09)	<0.001	2.87	(2.12–3.87)	<0.001	<0.001	2.05	(1.40–3.00)	<0.001	2.74	(1.90–3.96)	<0.001	<0.001
Shoulders			1.93	(1.35–2.77)	<0.001	2.37	(1.72–3.25)	<0.001	<0.001	1.83	(1.22–2.75)	0.004	2.32	(1.58–3.42)	<0.001	<0.001
Back			1.44	(1.02–2.03)	0.04	1.48	(1.10–2.00)	0.01	0.01	1.23	(0.84–1.80)	0.29	1.36	(0.94–1.96)	0.10	0.10
Elbows			1.01	(0.70–1.48)	0.94	1.03	(0.74–1.42)	0.87	0.86	1.17	(0.76–1.79)	0.47	1.25	(0.83–1.88)	0.28	0.28
Lower back or waist			1.15	(0.82–1.62)	0.41	0.98	(0.73–1.32)	0.88	0.91	1.25	(0.85–1.83)	0.25	1.21	(0.85–1.74)	0.29	0.30
Wrists or hands			1.14	(0.81–1.61)	0.44	1.11	(0.83–1.49)	0.48	0.47	1.16	(0.80–1.70)	0.44	1.17	(0.82–1.69)	0.38	0.39
Hips or legs			0.93	(0.62–1.40)	0.73	1.08	(0.77–1.53)	0.66	0.67	0.97	(0.62–1.54)	0.91	1.28	(0.83–1.98)	0.26	0.25
**Visual system**																
Eye discomfort (severe) (Ref: never/mild-moderate)			1.42	(0.80–2.51)	0.24	4.00	(2.57–6.22)	<0.001	<0.001	1.43	(0.76–2.70)	0.27	4.07	(2.35–7.06)	<0.001	<0.001
Headache (severe) (Ref: never/mild-moderate)			1.36	(0.51–3.64)	0.53	2.85	(1.34–6.07)	0.01	0.01	1.37	(0.47–3.98)	0.57	3.05	(1.20–7.77)	0.02	0.01
CVS (Ref: never/mild-moderate)			1.00	(0.62–1.63)	0.99	0.91	(0.61–1.36)	0.64	0.63	1.07	(0.62–1.86)	0.80	1.08	(0.65–1.77)	0.77	0.79
**Mental health condition**																
BSRS-5 ≥6 (Ref: BSRS-5 < 6)			1.29	(0.21–7.79)	0.78	1.58	(0.35–7.13)	0.55	0.55	1.02	(0.12–8.31)	0.99	2.23	(0.30–16.49)	0.43	0.40
Burnout (Ref: no)			0.60	(0.28–1.30)	0.20	1.02	(0.58–1.79)	0.96	0.99	0.60	(0.26–1.39)	0.23	1.35	(0.66–2.74)	0.41	0.42
Job stress (Ref: mild/ moderate)			1.09	(0.64–1.85)	0.75	1.65	(1.08–2.53)	0.02	0.02	1.16	(0.64–2.11)	0.62	2.29	(1.32–3.99)	0.003	0.003
**SRH**																
Self-rated Health Status (poor) (Ref: good/moderate)			0.88	(0.42–1.83)	0.73	1.20	(0.67–2.15)	0.55	0.56	0.90	(0.40–2.01)	0.79	1.71	(0.81–3.58)	0.16	0.16
Sleep duration <6 h (Ref: ≥6 h)			1.11	(0.66–1.85)	0.70	1.18	(0.76–1.83)	0.46	0.45	1.06	(0.60–1.89)	0.83	1.33	(0.77–2.32)	0.31	0.31
Self-assessment of sleep quality (poor) (Ref: good)			1.52	(1.08–2.14)	0.02	1.52	(1.14–2.04)	0.004	0.004	1.40	(0.96–2.06)	0.08	1.53	(1.06–2.20)	0.02	0.02
**b. Doctor/nurse group**																
**Variables**			**Model 1**							**Model 2**						
			**2–6 h**			**≥6 h**				**2–6 h**			**≥6 h**			
			**aOR**	**95% CI**	***p* value**	**aOR**	**95% CI**	***p* value**	***p* trend**	**aOR**	**95% CI**	***p* value**	**OR**	**95% CI**	***p* value**	***p* trend**
**Musculoskeletal system**																
Neck			1.15	(0.88–1.50)	0.31	1.73	(1.37–2.17)	<0.001	<0.001	0.93	(0.68–1.28)	0.67	1.30	(1.02–1.74)	0.01	0.02
Shoulders			1.57	(1.18–2.09)	0.002	2.67	(2.09–3.42)	<0.001	<0.001	1.22	(0.87–1.72)	0.24	1.90	(1.39–2.59)	<0.001	<0.001
Back			1.53	(1.17–2.01)	0.002	1.93	(1.54–2.43)	<0.001	0.01	1.23	(0.90–1.70)	0.19	1.49	(1.12–1.98)	0.01	0.004
Elbows			2.04	(1.45–2.86)	<0.001	1.97	(1.47–2.65)	<0.001	<0.001	1.62	(1.09–2.43)	0.02	1.55	(1.08–2.24)	0.02	0.06
Lower back or waist			1.45	(1.11–1.89)	0.01	1.86	(1.49–2.32)	<0.001	<0.001	1.29	(0.94–1.76)	0.12	1.58	(1.20–2.10)	0.001	0.001
Wrists or hands			1.69	(1.28–2.24)	<0.001	1.94	(1.53–2.45)	<0.001	<0.001	1.41	(1.02–1.95)	0.04	1.59	(1.19–2.12)	0.002	0.003
Hips or legs			1.29	(0.95–1.76)	0.11	1.60	(1.24–2.06)	<0.001	<0.001	1.18	(0.82–1.69)	0.37	1.45	(1.05–2.00)	0.02	0.01
**Visual system**																
Eye discomfort (Ref: never/mild/moderate)			1.32	(0.84–2.09)	0.23	3.62	(2.53–5.19)	<0.001	<0.001	0.97	(0.58–1.63)	0.90	2.49	(1.61–3.84)	<0.001	<0.001
Headache (Ref: never/mild/moderate)			1.33	(0.81–2.20)	0.26	2.26	(1.51–3.38)	<0.001	<0.001	1.01	(0.57–1.80)	0.97	1.57	(1.12–2.59)	0.04	0.02
CVS (Ref: never/mild/moderate)			1.99	(1.35–2.92)	<0.001	1.89	(1.39–2.58)	<0.001	<0.001	1.34	(0.84–2.11)	0.22	1.26	(0.85–1.88)	0.25	0.37
**Mental health condition**																
BSRS-5 ≥6 (Ref: BSRS-5 < 6)			0.65	(0.72–1.54)	0.33	1.35	(0.73–2.48)	0.34	0.21	0.94	(0.35–2.56)	0.90	1.65	(0.75–3.63)	0.21	0.12
Burnout (Ref: no)			1.30	(0.71–2.40)	0.40	2.07	(1.26–3.39)	0.004	0.002	1.36	(0.67–2.75)	0.40	2.08	(1.13–3.84)	0.02	0.01
job stress (severe) (Ref: mild/ moderate)			1.19	(0.87–1.61)	0.27	2.38	(1.86–3.04)	<0.001	<0.001	1.19	(0.82–1.71)	0.36	2.25	(1.63–3.10)	<0.001	<0.001
**SRH**																
Self-assessment of health status (poor) (Ref: good/moderate)			1.37	(0.84–2.22)	0.21	2.29	(1.55–3.39)	<0.001	<0.001	2.12	(1.20–3.74)	0.01	3.42	(2.07–5.64)	<0.001	<0.001
Sleep duration <6 h (Ref: ≥6 h)			1.74	(1.17–2.58)	0.01	2.05	(1.46–2.86)	<0.001	<0.001	1.42	(0.90–2.26)	0.14	1.47	(0.97–2.23)	0.07	0.10
Self-assessment of sleep time (poor) (Ref: good)			0.89	(0.68–1.16)	0.38	1.41	(1.13–1.76)	0.002	0.001	0.88	(0.64–1.21)	0.44	1.30	(1.03–2.23)	0.03	0.01

Note: VDT time: reference <2 h. Model 1: unadjusted. Model 2: adjusted for age, sex, smoking, drink, exercise, seniority, working hours and working style. CI = confidence interval, aOR = adjusted odds ratio. Other abbreviations are as show in Table 2.

**Table 4 ijerph-19-03770-t004:** Adjusted odds ratio (aOR) and 95% CI of the musculoskeletal system, visual system, mental health condition, and self-rated health status according to different durations of daily VDT working time using logistic regression analysis stratified by age and sex.

Subgroup	Number of Pain Areas ^a^			CVS ^a^			BSRS-5 ^a^			Burnout ^a^			SRH ^a^		
	OR	(95% CI)	*p* Value	OR	(95% CI)	*p* Value	OR	(95% CI)	*p* Value	OR	(95% CI)	*p* Value	OR	(95% CI)	*p* Value
Male < 30 y (*n* = 74)															
2–6 h	0	0	1.00	0	0	1.00	1.10	(0.29–4.23)	0.89	3.00	(0.44–20.38)	0.26	2.09	(0.17–25.19)	0.56
≥6 h	1.83	(0.30–11.24)	0.51	1.77	(0.15–21.09)	0.65	2.20	(0.65–7.40)	0.20	0	0	1.00	3.83	(0.49–30.09)	0.20
**Male ≥ 30 y (*n* = 244)**															
2–6 h	2.17	(0.91–5.15)	0.08	0.48	(0.05–4.20)	0.51	0.91	(0.41–1.99)	0.81	0.80	(0.21–3.09)	0.75	0.73	(0.23–2.36)	0.60
≥6 h	1.30	(0.53–3.18)	0.57	2.94	(1.09–3.98)	0.004	0.93	(0.45–1.91)	0.84	1.09	(0.35–3.40)	0.88	1.06	(0.40–2.80)	0.91
**Female < 30 y (*n* = 855)**															
2–6 h	1.52	(0.86–2.68)	0.15	0.96	(0.51–1.78)	0.89	0.89	(0.62–1.30)	0.55	1.50	(0.61–3.70)	0.37	1.48	(0.79–2.79)	0.22
≥6 h	2.18	(1.38–3.47)	0.001	2.00	(1.27–3.14)	0.003	1.23	(1.01–3.26)	0.03	2.29	(1.12–4.70)	0.02	2.63	(1.60–4.33)	<0.001
**Female ≥ 30 y (*n* = 1640)**															
2–6 h	1.27	(0.90–1.80)	0.17	1.65	(1.02–2.67)	0.04	1.88	(1.37–2.58)	<0.001	0.70	(0.37–1.30)	0.26	1.40	(0.75–2.59)	0.29
≥6 h	1.06	(0.79–1.43)	0.68	4.60	(3.10–6.74)	<0.001	2.25	(1.71–2.95)	<0.001	1.33	(0.83–2.12)	0.23	2.35	(1.41–3.91)	0.001

VDTs working time: reference < 2 h. ^a^ Number of pain area: ≥5 vs. <5. ^b^ CVS: yes vs. no. ^c^ BSRS-5: ≥6 vs. <6. ^d^ Burnout: yes vs. no. ^e^ SRH: Self-assessment of health status (poor) + self-assessment of sleep quality (poor) vs. Self-assessment of health status (good or moderate) or self-assessment of sleep time (good).

## Data Availability

No data are available.

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
