# Peer review of "Influence of Prolonged Visual Display Terminal Use on Physical and Mental Conditions among Health Care Workers at Tertiary Hospitals, Taiwan"

_ijerph, 2022, doi:10.3390/ijerph19073770_

Round 1
Reviewer 1 Report
The title of the article is very interesting to me because of the "metal health" and VDT. There are a lot of studies on physical problems, but the metal health. However, there is not much about it in detail. It would be good if the author can discuss more about metal health.
Author Response
Response to Reviewer 1 Comments
Many thanks for your helpful comments. In response to these comments we have revised the manuscript as follow:
Point: The title of the article is very interesting to me because of the "metal health" and VDT. There are a lot of studies on physical problems, but the metal health. However, there is not much about it in detail. It would be good if the author can discuss more about metal health.
Response: Thanks for the reviewer’s recommendation and reminder.
We have added the discussion of VDT and mental health, including anxiety, depression, insomnia, and burnout.
“Previous studies mentioned that visual fatigue was significantly associated with anxiety perception in a dose-related matter (OR 7.40; 95% CI 1.77–31.3), psychosocial factors (OR 1.03; 95% CI 1.01–1.07), and VDT usage time (OR 1.27; CI 95% 1.04–1.53) [31]. A systematic review and meta-analysis of studies that compared those who reported less and more screen time-based sedentary behavior (ST-SB), showed that persons reporting more ST-SB had a significantly higher risk of depression, especially in female groups (pooled OR: 1.18) [32, 33]. More time spent using a computer on weekdays was associated with an increased risk of anxiety and depression [34]. A converse view in another study showed that the evidence for anxiety symptoms was more limited than that for depressive symptoms in adults and were therefore more inconsistent [33]. According to self-reported depressive and anxiety disorder in a nationally representative sample of South Korean workers, after adjustment factors, the OR for depressive and anxiety disorder was higher in workers using computers more than 75 % of their workday (OR 1.69, 95 % CI 1.30−2.20) than in workers using computers less than 50 % of their shift [23]. There was a significant that burnout appeared to be a significant contributing factor to the intensity and frequency of VSD [35].
Giahi et al. collected data from 382 Iranian bank tellers, and found significantly higher insomnia symptom scores, higher job stress, and lower job satisfaction among participants with more than six hours of daily VDT use after adjusting for multiple confounding factors (P < 0.001) [36]. Yoshioka et al. investigated 2,417 Japanese clerks (men: 2,030; women: 387) at local government offices. They found that VDT work of six hours or more per day was significantly associated with insomnia (OR = 1.62; 95% CI: 1.16–2.27), even after adjusting for possible confounding factors [37].” (Line 431-452).
Thank you again for your constructive comments.
Reviewer 2 Report
Thank you for opportunity to review this manuscript.
This study deals with the effects of prolonged VDT working time on physical and mental health disadvantages among health care workers (HCWs). However there are many issues that need to be adressed.
1. There is a difference in the number of non-doctor/nurse and doctor/nurse groups. (945 vs 1868) This should be statistically corrected. (Propensity score matching method)
2. page 5, 6. Table 1, 2, 3: What is p trend?
3. Table 1: What is the criteria for dividing working hours/week? (≦ 45hrs, 46-50hrs, 51-59hrs, ≧60 hrs)
4. Table 4: What is the basis for classifying age by age 30?
5. Discussion
There is a need to updating the bibliography in the article.
6. Conclusion
The authors need to provide detailed description about implication for practice and research.
Author Response
Response to Reviewer 2 Comments
Many thanks for your helpful comments. In response to these comments we have revised the manuscript as follow:
Point 1: There is a difference in the number of non-doctor/nurse and doctor/nurse groups. (945 vs 1868) This should be statistically corrected. (Propensity score matching method).
Response 1: Thanks for the reviewer’s recommendation and reminder.
According to this study, the use of computers by doctor/nurse and non-doctor/nurse groups were described separately (Table 3 a, b) (Page 12-14)
In principle, the two groups were not needed to compare with each other, and the method should not need to use the Propensity score matching method.
In addition, if this Propensity score matching method was used to match, there would be only 945 pairs (1890 people) left, and the 2533 people who could not meet the power force will be rejected (Line 80-82).
Point 2: Page 5, 6. Table 1, 2, 3: What is p trend?
Response 2: Thanks for the reviewer’s recommendation and reminder.
We have added the information.
“Cochran-Armitage trend test”.
Point 3: Table 1: What is the criteria for dividing working hours/week? (≦ 45hrs, 46-50hrs, 51-59hrs, ≧60 hrs).
Response 3: Thanks for the reviewer’s recommendation and reminder.
We have added the information.
“According to the Labor Standards Law in Taiwan, the "Working Hours Standards for Nursing Staffs in Medical Institutions,": the regular working hours per day shall not exceed eight hours. Furthermore, the total working hours per week shall not exceed 48 hours [22]. Regarding national regulations and South Korean research, we divide weekly working hours into four grades: ≤45 hours, 46-50 hours, 51-60 hours, ≧61 hours [23].” (Line 106-111).
Point 4: Table 4: What is the basis for classifying age by age 30?
Response 4: Thanks for the reviewer’s recommendation and reminder.
“We grouped according to median age: 45 years in the non-doctor/nurse group, 28 years in the doctor/nurse group, 32 years in all groups, and 30 years as the cut-off point for the estimated stratification.” (Line 185-187).
Discussion
Point 5: There is a need to updating the bibliography in the article.
Response 5: Thanks for the reviewer’s recommendation and reminder.
We have updated the references. (Page 20-22)
Conclusion
Point 6: The authors need to provide detailed description about implication for practice and research.
Response 6: Thanks for the reviewer’s recommendation and reminder.
We have added the description for practice and research.
“The most effective way to eliminate MSDs, CVS hazards, and mental health problems in the tertiary hospital workplace is to develop health programs. The program aimed at three fields: first, advocating healthy lifestyle behaviors (stretching exercise, quitting smoking, and adequate sleep time/quality) [23, 42]. Second, raise workers’ awareness of ergonomics of workstation ( seat-plate height adjustment, working desktop height, key-board and mouse arrangement, screen contrast brightness, and sight distance). Third, work organization (VTD working duration, rotation and shift job, job stress, and social support) [23, 42, 43], especially for women and younger workers.” (Line 470-477)
Thank you again for your constructive comments.
Reviewer 3 Report
Thank you for the opportunity to review the manuscript “Influence of Prolonged Visual Displayer Terminal Use on Physical and Mental Conditions among Health Care Workers at Tertiary Hospitals, Taiwan”. Congratulations to the author for his work. This study is very interesting to assess the impact on workers’ physical and mental health of time spent in front of a videoterminals (vdts). In the study, the author highlights the repercussions of many variables, first of all being in the medical/nursing group or in the non-medical/nursing group and the time of exposure to VDTs, on quality of life, working life and consequently on the quality of the service provided to patients. Finally, the authors highlight the high costs derived from the consequent absenteism and lost work productivity.
Introduction is good and well explained.
Materials and methods described the characteristics of this cross-sectional study and groups are well selected and based on job categories. PROS are the division on two groups, the use of Nordic Musculoskeletal Questionnaire (NMQ) for the evaluation of physical ilness and the MBI-HSS for the work-related stress.
The results are well detailed, with supplementary graphs and tables, and explain attitudes according to socio-demographic variables. The use of this tables makes the understanding of results clear.
The discussion is comprehensive and takes up what has been published in the previous literature on this topic.
Conclusions are just focused on workplace risk assessment and highlights what needs to be done to prevent MSDs, visual symptoms and ensure good mental health.
I agree with the limitations of this study, as stated by the author, mainly on the significant difference in sample size between men and women. The results cannot be generalised to other health care workers because the workers in this study were recruited from only two hospitals.
Author Response
XCV
Response to Reviewer 3 Comments
Many thanks for your helpful comments. In response to these comments we have revised the manuscript as follow:
Point 1: Thank you for the opportunity to review the manuscript “Influence of Prolonged Visual Displayer Terminal Use on Physical and Mental Conditions among Health Care Workers at Tertiary Hospitals, Taiwan”. Congratulations to the author for his work. This study is very interesting to assess the impact on workers’ physical and mental health of time spent in front of a videoterminals (vdts). In the study, the author highlights the repercussions of many variables, first of all being in the medical/nursing group or in the non-medical/nursing group and the time of exposure to VDTs, on quality of life, working life and consequently on the quality of the service provided to patients. Finally, the authors highlight the high costs derived from the consequent absenteism and lost work productivity.
Response 1: Thanks for the reviewer’s comments.
Introduction
Point 2: Introduction is good and well explained.
Response 2: Thanks for the reviewer’s comments.
Materials and Methods
Point 3: Materials and methods described the characteristics of this cross-sectional study and groups are well selected and based on job categories. PROS are the division on two groups, the use of Nordic Musculoskeletal Questionnaire (NMQ) for the evaluation of physical ilness and the MBI-HSS for the work-related stress.
Response 3: Thanks for the reviewer’s comments.
Results
Point 4: The results are well detailed, with supplementary graphs and tables, and explain attitudes according to socio-demographic variables. The use of this tables makes the understanding of results clear.
Response 4: Thanks for the reviewer’s comments.
Discussion
Point 5: The discussion is comprehensive and takes up what has been published in the previous literature on this topic.
Response 5: Thanks for the reviewer’s comments.
Conclusion
Point 6: Conclusions are just focused on workplace risk assessment and highlights what needs to be done to prevent MSDs, visual symptoms and ensure good mental health. Response 6: Thanks for the reviewer’s recommendation and reminder.
We have added the description for workplace risk assessment and prevention.
“The most effective way to eliminate MSDs, CVS hazards, and mental health problems in the tertiary hospital workplace is to develop health programs. The program aimed at three fields: first, advocating healthy lifestyle behaviors (stretching exercise, quitting smoking, and adequate sleep time/quality) [23, 42]. Second, raise workers’ awareness of ergonomics of workstation ( seat-plate height adjustment, working desktop height, key-board and mouse arrangement, screen contrast brightness, and sight distance). Third, work organization (VTD working duration, rotation and shift job, job stress, and social support) [23, 42, 43], especially for women and younger workers.” (Line 470-477)
Limitation
Point 7: I agree with the limitations of this study, as stated by the author, mainly on the significant difference in sample size between men and women. The results cannot be generalised to other health care workers because the workers in this study were recruited from only two hospitals.
Response 7: Thanks for the reviewer’s comments.
Thank you again for your constructive comments.